# Gender Assignment to Spanish Pseudowords by Monolingual and Basque-Spanish Bilingual Children

**Rocío Pérez-Tattam [1],\*, Maria José Ezeizabarrena [2],\*, Hans Stadthagen-González [3]
and Virginia C. Mueller Gathercole [4]**

[1]  Modern Languages, Translation and Interpreting, Swansea University, Swansea SA2 8PP, Wales, UK
[2]  Linguistics and Basque Studies, Faculty of Arts, University of the Basque Country (UPV/EHU),
    01006 Vitoria-Gasteiz, Spain
[3]  School of Psychology, The University of Southern Mississippi, Long Beach, MS 39560, USA
[4]  Linguistics Programme, Florida International University, Miami, FL 33199, USA
**\***  Correspondence: r.s.perez-tattam@swansea.ac.uk (R.P.-T.); mj.ezeizabarrena@ehu.eus (M.J.E.)

**Abstract:** This study examines gender marking in the Spanish of Basque-Spanish bilingual children. We analyze data collected via a production task designed to elicit 48 DPs, controlling for gender of referents and for number and types of morphological cues to grammatical gender. The goals were to determine the extent to which participants rely on biological cues (female referent =>FEM gender, male referent =>MASC gender) and morpho-phonological cues (*-a* ending =>FEM, *-o* ending =>MASC, others =>MASC or FEM) to assign gender to pseudowords/novel words; and whether bilinguals' language dominance (Spanish strong/weak) has an effect. Data were collected from 49 5- to 6-year-old Spanish-speaking children—28 monolingual L1 Spanish (L1$_{Sp}$) and 21 Basque-dominant (L1 Basque-L2 Spanish) bilinguals (BDB). Results reveal a general preference for MASC gender across conditions, especially in BDB children, who produced masculine modifiers for 83% of items, while the L1$_{Sp}$ children did so for only 63% of items. Regression analyses show that for both groups, morphological cues have more weight than the nature of the referent in participants' assignment of gender to novel words, and that the L1$_{Sp}$ group is more attentive to FEM morphological markers than the BDB group, pointing towards the existence of differences in the strength of cue-patterns for gender marking.

**Keywords:** gender; early bilinguals; Basque-Spanish; Pseudowords

## 1. Introduction

Grammatical gender is a nominal feature coded in the lexicon and/or the morphology of many, though not all, languages. However, noticeable differences are found across languages with grammatical gender, depending on whether it is overtly coded in the nominal domain (Romance languages) and/or in the verbal domain (Basque, Hebrew), and in the specific categories bearing the feature (nouns (Ns), pronouns, determiners (Dets), adjectives, participles, or verb (V) inflections), and in the morphological marking (suffix/prefix/suppletive forms). Moreover, gender marking varies from more transparent and regular morphology, as in Spanish, Italian, or Greek, to more opaque or irregular morphology, such as in the case of languages like German and Dutch. Additionally, the number of gender classes varies from languages distinguishing two classes, MASC(uline) and FEM(inine)[1], like Italian, French and Spanish, to languages distinguishing three (MASC, FEM, Neuter), like German and Greek, or even four, like Polish (Corbett 1991).

---

[1]   MASC and FEM will be used throughout to indicate linguistic gender, masculine and feminine.

*1.1. Gender with Real and Non-Real Words*

The research on language acquisition has shown that L1 and L2 learners are sensitive to this grammatical feature from very early on (Hernández-Pina 1984; Larrañaga and Guijarro-Fuentes 2013). Nevertheless, many types of errors during the early years of language use are attested across studies in the spontaneous production of monolingual and bilingual learners; these show that gender errors are less persistent in early L1 than in late L2 (Carroll 1989; Gathercole 2002; Gathercole and Thomas 2009; Meisel 2018) and in monolingual than in bilingual language acquisition (Cuza and Pérez-Tattam 2015; Larrañaga and Guijarro-Fuentes 2013). Inter-individual differences between early bilinguals who have acquired the gender language simultaneously versus successively have also been found in experimental production studies, which leads to the conclusion that age as well as the amount of exposure during the earliest ages affect individuals' acquisition of such features (Unsworth et al. 2014).

There is debate on whether bilinguals can fully acquire both languages, or even one. Age of Acquisition (AoA), together with the amount of exposure to each of the languages, seems to play a role in the degree of competence (fluency), but inter-individual differences have been found even among bilinguals who have acquired both languages during their childhood. Sociolinguistic dominance also affects bilingual individuals' competence. For instance, bilinguals with different degrees of exposure to their respective languages during childhood may develop a similar command of the sociolinguistically dominant language, regardless of their exposure to that language at home, whilst these same bilinguals' competence in the minority language will be more varied across individuals, since it will be more related to the intensity, use domains, and quality of the input for each participant from the minority language (Gathercole 2016; Gathercole and Thomas 2009). Thus, processing studies conducted with early sequential and simultaneous bilinguals with two gender languages such as Russian and German have revealed different gender processing patterns depending on whether children have acquired both languages simultaneously or sequentially: sequential bilingual children, in contrast to simultaneous bilinguals, may have access to L2 nouns through the L1 lexicon (Lemmerth and Hopp 2017).

Many grammatical gender studies on child and adult L1 and L2 are based on real words (Brisk 1976; Lemmerth and Hopp 2017; Lew-Williams and Fernald 2008). Nevertheless, some have probed children's knowledge of grammatical gender using novel words (Arias-Trejo and Alva 2013; Arias-Trejo et al. 2013; Karmiloff-Smith 1978; Pérez-Pereira 1991). Arias-Trejo and colleagues conducted a multimodal visual-auditory experimental task with Mexican infants and showed that children are sensitive to the morpho-syntactic markers of gender in Spanish: a very short exposure to novel words with -*o*/-*a* endings (*pileco*/*betusa*) created to refer to inanimate objects may be enough for even 30-month-olds to associate novel words with MASC/FEM gender features. Data revealed that children paid attention not only to word endings but also to the endings of adjectives and determiners in nounless determiner phrases (DPs) used in the experiment to refer to referents. In fact, children were able to anticipate the target image based on the gender marking of the adjective (*es amarillo/-a*; *es bonito/-a* 'it is yellow-MASC/FEM; it is pretty-MASC/FEM') or the Det (*la*/*el*) produced before (and without mentioning) the novel word. These results led researchers to conclude that children are able to associate novel words with a specific gender based on the morpho-syntactic cues of nouns, adjectives and determiners (Arias-Trejo and Alva 2013; Arias-Trejo et al. 2013). Similarly, studies with older children confirm Spanish-speaking children's sensitivity to word endings as a reliable cue for gender assignment to novel words (Ogneva 2019).

In contexts of real words, it is not always easy to disentangle the source of participants' good or poor performance. For example, participants' target-deviant responses may be related to any of the following causes:

(a)    a lexical deficit associated with the semantic properties or with an incomplete representation of the phonological form of the specific items under study,

(b)    a morphological deficit in the knowledge of the inflection paradigms,

(c)    a syntactic deficit in the processes governing the local dependencies between the N and its modifiers (determiners, adjectives),

(d)   a general limitation in the area of working memory or some immature or limited lexical access or in the resources required to process linguistic information.

Deficits of type (a) may be especially affected by the frequency of the lexical items in the language, in general, and by participants' degree of familiarity with them. In contrast, experiments using non-real words allow the teasing apart of the lexical properties (inherent gender features of each N) from the morpho-syntactic processes related to them (grammatical features) as well as from non-linguistic factors affecting participants' responses, such as frequency, memory, and so on.

The use of non-real words in experiments allows one to control these various factors. The use of novel nouns may be especially revealing in relation to child language research for two reasons: (a) early L1 acquisition studies conclude that learning the grammatical gender features associated with each lexical item requires more time (in years) than learning other formal (number, case marking) and semantic properties (animacy, size) associated with nominal categories (N, Adj) in gender languages, and (b) information on the frequency and productivity of lexical items in the language is often based on written and formal corpora, and it is, consequently, difficult to evaluate the effects of these features of particular lexical items on children's productive use. Additionally, (c) experimental tasks based on non-real words allow control for other factors associated with the phonological forms and the semantic properties of words.

The question that arises when considering all of the factors related to grammatical gender is whether there is a hierarchy in the reliance on available cues to gender when assigning gender to a novel word. We specifically ask to what extent 5-year-old children determine a given noun's gender, and to what extent they rely on the following factors in making that determination: (a) the semantic class associated with sex such as animate masculine, animate feminine, inanimate referents (artifacts), (b) the N's phonological form, such as its ending, and (c) the distributional privileges of occurrence with nominal modifiers. All of these may correlate highly with grammatical gender in grammatical gender languages, so they may be employed by a user of the language to predict or determine the gender of a novel noun.

## 1.2. Strategies for Gender Marking

In gender languages like Spanish where Ns are assumed to bear inherent (interpretable) gender features, and Det and Adj to bear non-interpretable gender features according to the N's features in Det __ Adj contexts, differences have been found in the strategies for gender marking of Det and Adj in monoglot Spanish speakers producing monoglot DPs and in bilinguals producing mixed DPs composed of a DetSp and a lexical insertion from a different language. For example, Basque-Spanish bilinguals may produce feminine monoglot DPs (*la miel* 'the-FEM honey') and a masculine (*el EZTI* or *el EZTIA* 'the-MASC honey') or a feminine mixed DP (*la EZTI* or *la EZTIA* 'the-FEM honey') (Munarriz et al. 2018) when inserting the lexical equivalent *ezti* 'honey' from Basque, a language in which nouns are not marked for gender.

The following strategies have been attested in the literature on pure and mixed DPs:

(a)   *morpho-phonological cues* such as an *-a* ending on a N are associated with FEM, whilst an *-o* ending on a N is associated with MASC (Harris 1991); other endings like *-n* or *-e* have some or no clear preference for one of them (see Teschner and Russell 1984 for gender frequencies in Spanish corpora, as illustrated by the MASC/FEM alternation (1a/2a and 1b/2b));

(b)   *analogical gender*, according to which the English word inserted in a mixed Spanish DP is assigned the gender corresponding to the Spanish equivalent word—e.g., *galleta* 'cookie-FEM' (1a), *libro* 'book-MASC'(1b);

(c)   (masculine or feminine) *default gender*, according to which an inserted English N tends to be assigned MASC gender (2a, 2b), whilst some Basque-Spanish bilinguals tend to assign FEM to an inserted Basque word ending in *-a*, such as *ariketa* and *idazlana* (2c, 2d) regardless of its analogical gender (*ariketa* = Sp 'ejercicio-MASC' and *idazlana* = Sp: 'redacción-FEM') (see Couto et al. 2016; Cuza and Pérez-Tattam 2015; Imaz 2016; Liceras et al. 2018; Liceras et al. 2008; Munarriz et al. 2018; Valdés Kroff et al. 2017).

(1)  a.  *la*          *COOKIE*        b.  *el*          *BOOK*
         the-FEM     cookie              the-MASC    book
(2)  a.  *el*          *COOKIE*        b.  *el*          *BOOK*
         the-MASC    cookie              the-MASC    book
     c.  *la*          *ARIKETA*       d.  *la*          *IDAZLANA*
         the-FEM     exercise            the-FEM     writing task

*1.3. Early Monolingual vs. Bilingual Grammars*

The assumption that input, in interaction with the age of onset (AO), models language acquisition processes is generally accepted. Consequently, L2 acquisition is considered to differ from L1 acquisition, as is the case for bilingual compared to monolingual language acquisition. Some researchers consider that L1 and L2 learners differ in their knowledge systems, since implicit learning driven by Universal Grammar (UG) may be possible only in L1, whereas L2 learners may have to make use of additional cognitive learning resources to compensate any gaps. More debated is the issue of whether early simultaneous bilinguals (2L1) and early L2 learners (sequential bilinguals) achieve the same ultimate grammatical knowledge as monolingual L1 learners and, more specifically, whether they go through the same developmental stages. An additional controversial issue is the AO at which consecutive L2 (cL2) learners reveal qualitative differences from monolingual (and bilingual) L1 learners. To our knowledge the amount of minimal input required for bilinguals (simultaneous or sequential) to acquire a native-like competence in general, and more particularly in the acquisition of grammatical features like gender, is not well established, but some suggest that the relevant boundary at which cL2 acquisition appears similar to L1 is age 3;6 (Meisel 2018).

Evidence has shown that early bilinguals may master gender marking at different ages in the acquisition process of two gender languages. An interesting case is that of Greek and Dutch bilinguals, who produce adult-like marking early in Greek (around age 3), but not until age 6 or so in Dutch (Unsworth et al. 2014). These authors concluded that the different degree of transparency in the two languages (Greek gender transparent, Dutch gender opaque) might explain such asynchrony. Moreover, they argued that cumulative input rather than AO explains differences between early simultaneous and successive Greek-Dutch bilinguals with differing amount of exposure.

Moreover, studies on (older) heritage speakers find differences between such bilinguals (early bilinguals with different exposure to their two languages) and monolinguals. The less accurate performance in morphosyntax in different types of unbalanced bilinguals has been proposed to result from *incomplete acquisition* (Montrul 2008) or *language attrition* (Polinsky 2011). More specifically, differences found in gender marking of real words between monolinguals and different types of bilinguals support the accounts mentioned above (Montrul and Potowski 2007; Polinsky 2008). Models focusing on the process of language acquisition rather than on the ultimate attainment propose a different view. According to the *Feature re-assembly Hypothesis* (Lardiere 2005; Putnam and Sánchez 2013) bilinguals' target-deviant performance in morphosyntax is due to a low(er) level of activation of language features (functional, phonological, and semantic features) in their less strong language, rather than to incomplete linguistic knowledge.

The present study examines the extent to which Basque dominant Basque-Spanish bilingual children (BDB) rely on the available cues to gender when learning and using novel nouns in Spanish, as compared to monolingual Spanish children. In other words, we aimed to determine:

(a)  the **cues** children **rely on** the most when assigning gender to novel words: semantic (biological gender) **cues** (female =>FEM, male =>MASC), morpho-phonological cues (*-a/-o*/other endings), or morpho-syntactic **cues** such as Spanish DP context ( Det ___, ___adj; Det ___ adj),

     and

(b)  the effect of **language dominance (strong vs. weak(er) L)** in the strategies for gender assignment.

Based on the literature reviewed the following predictions can be made:

1)　All participants who have already acquired the Spanish gender rules (5-year-olds with regular exposure to Spanish) will show that

    a.　both semantic (referential) and

    b.　formal (morpho-syntactic and morpho-phonological) cues will drive children's gender assignment to novel words (pseudowords);

2)　5-year-old BDB children will differ from their L1$_{Sp}$ monolingual peers in the strategies for gender assignment. In particular, the bilinguals will show **less consistency**

    a.　in the (strength of) cues for gender assignment to N (lexicon) and

    b.　in the (regularity) of (Det)-(N)-(Adj) phrase-internal agreement: (morphosyntax).

## 2. The Study

The design of this study was adapted from studies developed by the Experimental-Developmental Research Group at the Economic and Social Research Council (ESRC) Centre for Bilingualism to examine grammatical gender knowledge in Welsh-English, Spanish-English, and Spanish-Welsh bilinguals. For both Welsh and Spanish, novel nouns were developed, as well as novel picture characters/objects as referents. The nouns were constructed in accordance with the morpho-syntactic properties of the particular language involved, and these were accompanied by modifiers (determiners and adjectives) that were or were not marked for gender. Referent objects were constructed as either animate or inanimate objects (artifacts), and the animate objects showed male characteristics or female characteristics or were non-specific with respect to gender of the referent. Productive use of gendered forms was elicited by providing zero or one or more gender cues (on the noun or in one or more modifiers) in one stimulus and eliciting a gendered form of another modifier as a response. By controlling carefully for nominal form, modifier markings, and referential properties for novel nouns in the prompt stimuli, it is possible to assess the role of each of these factors in children's and adults' grammatical gender assignment and their application of that knowledge in producing a new form.

The research on Welsh-English bilinguals was reported in Sharp (2012). In the case of Welsh, the nominal morphology was controlled primarily via noun mutation, the primary marker for grammatical gender in Welsh (Gathercole et al. 2001), but gendered word endings (e.g., *-wr* ("-er, -or," as in *garddwr* 'gardener' (MASC) or *myfyriwr* student (MASC)) and *–es* ("-ess," as in *dynes* 'woman' (FEM) or *tywysoges* 'princess'(FEM))) were also included; and gendered modifiers involved gender-marked numerals for *two*, *three*, and *four*. Data from Welsh-English adults revealed that they made use of both gender endings and the sex of referents, in that conditions in which feminine endings and/or female referents occurred elicited more responses of feminine numerals than other conditions. Data from Welsh-English 5-, 7-, and 9-year-olds revealed limited use of feminine numerals at all ages; at age 9, a few children used the feminine numerals, either in a non-discriminatory fashion or according to the sex of the referent. The results for Welsh are largely attributable to the restricted use of gender marking in numerals in Welsh, limited to only a few numbers, and the highly complex nature of gender marking in Welsh.

Research on Spanish-English bilinguals in Miami reveal a slightly different picture of productivity with these forms. In that study (in prep), bilingual children from three types of language backgrounds (US-born English with Spanish at home, US-born with only Spanish at home, early immigrants with L1 Spanish-L2 English) were studied, and for whom the authors had receptive vocabulary scores in both English and Spanish. In the case of Spanish, nominal morphology was controlled via noun endings, with novel nouns constructed to end in *-o*, usually a marker for masculine gender, in *-a*, usually marking feminine gender, or *-e*, non-specific for gender; gendered modifiers involved gender-marked determiners and/or gender-marked adjectives. Data from the Miami bilinguals indicate that all factors influence response patterns—noun gender marking, the number of distributional cues for gender, and the sex of the referent were influential in performance. First, there is a tendency for the sex of the referent to be influential in participants' choice of

grammatical gender in responses, but this is especially true for those whose Spanish level is weaker, as indicated by lower vocabularies, or whose English level was higher, as indicated from being from English and Spanish at home (ESH) homes or having higher English vocabularies. Those with strong command of Spanish are more likely to produce grammatical gender in their responses that matched the grammatical gender of the input stimulus, irrespective of the sex of the referent. In general, masculine nouns elicited masculine responses, and feminine nouns feminine responses, but the more frequently cues were available in the modifiers in the experimental stimuli, the greater the number of responses in concord with the modifier gender; and the gender of referents was influential, especially in relation to linguistic stimuli that carried fewer linguistic cues to gender. The bilinguals' proficiency in English and Spanish affected results, however: those with lower abilities in Spanish, as judged either by their relatively higher scores on English vocabulary or their relatively lower scores on Spanish vocabulary, relied more heavily on the sex of referents, even when a high number of morphological gender cues were available in the prompt stimulus, than their peers who had higher proficiency in Spanish.

In Miami, which is a thriving multi-lingual city in which Spanish figures prominently, the dominant language is nevertheless English (Gathercole and Thomas 2009). One question is whether performance on grammatical gender in Spanish in these bilingual children's speech is affected by this fact, reflecting a level of incomplete learning of the less-dominant language. The dominant language, English, which only has natural gender, not grammatical gender, may be informing bilingual learning children's construction of grammatical gender in Spanish. An interesting comparison group would be to examine the acquisition of Spanish grammatical gender in bilingual children whose other language, like English, does not contain grammatical gender, but for whom the dominant community language is Spanish. The Basque-Spanish community in the Basque country provides such a comparison group. This study was conducted to examine Basque-Spanish-speaking children's performance on a similar task for Spanish grammatical gender.

*2.1. Experimental Design*

We looked at gender marking in novel nouns via an elicited production task. We elicited DPs (determiner + (novel noun) + adjective) in Spanish.

The elicitation of linguistic stimuli occurred in relation to a set of pictures involving novel referents, one of which moved via animation, and the participant had to indicate verbally which object had moved.

2.1.1. Linguistic Stimuli

DP INTERNAL AGREEMENT MARKERS. DETERMINERS AND ADJECTIVES. In Spanish, determiners and some adjectives have different forms or endings: masculine nouns combine with the forms of the definite article *el* (MASC sg.)/*los* (MASC pl.) and feminine nouns combine with the forms of the definite article *la* (FEM sg.)/*las* (FEM pl.). Overt marking of gender is also found in indefinite articles and demonstratives. Adjectives ending in *-o* combine with masculine nouns (3a), adjectives ending with *-a* combine with feminine nouns (3b).

(3)　a.　*el*　　　　*rey*　　　*roj-o*
　　　　　the-MASC　　king　　red-MASC
　　　　　'the red king'

　　b.　*la*　　　　*silla*　　*roj-a*
　　　　　the-FEM　chair　　　　red-FEM
　　　　　'the red chair'

We elicited definite articles, either the masculine form **el** (MASC sg.) or the feminine form **la** (FEM sg.), and gendered color adjectives: *amarillo/amarilla* 'yellow', *blanco/blanca* 'white', *negro/negra* 'black', *rojo/roja* 'red'.

We created audio-visual stimuli using distinct novel nouns that conformed to the morpho-phonotactic rules of Spanish. These stimuli were divided into four sets of 48 nouns, controlling for noun stem, noun ending, morpho-syntactic context and natural gender/animacy of the referent, as follows.

NOUN STEMS. The 48 stems were balanced for number of syllables (approximately half were monosyllabic, half were disyllabic). The stems are listed in Table 1.

**Table 1.** List of word stems used for building the pseudonouns.

| Word Stems | | | |
|---|---|---|---|
| Amitr- | Esont- | Melp- | Patibr- |
| Artald- | Filc- | Miob- | Rand- |
| Barent- | Golint- | Muest- | Rilap- |
| Biom- | Gorfil- | Nald- | Rist- |
| Blic- | Gurl- | Ners- | Rust- |
| Bratis- | Gurt- | Nert- | Senut- |
| Burant- | Licet- | Nolet- | Sut- |
| Calt- | Lindag- | Orent- | Tilen- |
| Cand- | Lipagr- | Ornet- | Tilent- |
| Chitr- | Lurc- | Ortib- | Tirb- |
| Delm- | Mardel- | Pald- | Tolc- |
| Elant- | Mel- | Pant- | Tuem- |

NOUN ENDINGS. We combined these stems with gendered endings (*-o* (MASC) and *-a* (FEM)) and the non-gendered ending *-e* as follows: *nolet-**o***, *nolet-**a***, *nolet-**e***. Half the novel nouns were presented in a context that favored masculine modifiers, half were presented in a context that favored feminine modifiers.

This created 192 word forms that were distributed across four different versions of the tests. These four versions of the task were balanced in that each version was assigned only one novel word with a given stem. That is, the four versions varied according to the endings of the stems, and to the modifier context, as shown in Table 2:

**Table 2.** Examples of forms built with the same root in the four versions.

| Version A | Version B | Version C | Version D |
|---|---|---|---|
| noleto | noleta | nolete | nolete |
| (MASC—context) | (FEM—context) | (MASC—context) | (FEM—context) |

The administration of each version was counterbalanced, with two different orders of presentation of the stimuli.

TARGET RESPONSES. Trials included two prompt sentences (experimenter), followed by one target sentence (participant). In each trial, the participant was asked to say the color of a novel object that was moving, combining the given noun with either the masculine or feminine form of the definite article and a gendered color adjective ending in *-o* or *-a*: **el**/**la** *(nolete) roj**o**/roj**a***.

PROMPTS. In each trial, the experimenter first showed a picture with exemplars of the novel object (as shown in Figure 1) and uttered two prompt sentences, in which the novel noun was presented. The novel noun was presented in one of three cue conditions: **zero cues** for grammatical gender of the noun, **one cue** (on the determiner or the adjective) or **two cues** (on the determiner and the adjective). For example, using the frame (4), the experimenter inserted one of the three options in (5): zero cues (5a), one cue (5b or 5c) or two cues (5d).

(4)    *Ahora vamos a ver….*
      'Now we are going to see…'
(5)   a.   *tres noletes*

　　　　　'Three noletes'
　　b.　　*unas noletes ∅*
　　　　　'some (FEM) noletes'
　　c.　　*∅ noletes simpáticas*
　　　　　'nice (FEM) noletes'
　　d.　　*unas noletes simpáticas*
　　　　　'some nice (FEM) noletes'

The experimenter then caused one of the objects to move (rotate, move up and down, or the like), and asked (second prompt) (6):

(6)　　*¿Cuál se está moviendo?* 'Which one is moving?'

### 2.1.2. Visual Stimuli

We used 48 base sets of four novel characters/objects novel pictures from the original studies to act as referents for the novel nouns, one set for each noun stem. Examples are shown in Figure 1. We then manipulated the attributes of each base picture to make it look (a) male, (b) female, (c) animate, but non-gendered as to biological gender, or (d) inanimate artifacts. Across the four versions of the task, a participant saw only one of the referents from each base picture set.

Across the four sets of stimuli, the administration of the linguistic stimuli was balanced for noun ending, morphological cues in the determiner and/or the adjective, and the non-linguistic stimuli were balanced for gender and animacy of the referents.

### 2.1.3. Procedure

Participants were seated in front of a computer screen and the experimenter gave them the following scenario:

*Imagina que ha llegado a la Tierra un platillo volante de un planeta lejano. Del platillo han salido muchas criaturas y objetos de ese planeta y han caído por todos lados. Ahora voy a enseñarte unos dibujos de estas criaturas y objetos, y voy a hacerte preguntas acerca de ellos.*

*Primero vamos a hacer unas de prueba con cosas que ya conoces.*

'Imagine that a flying saucer from a distant planet has arrived on Earth. Many creatures and objects from that planet have come out of the flying saucer and have landed everywhere. Now I'm going to show you some pictures of these creatures and objects, and I'm going to ask you questions about them.

First, we'll have a try with things you already know.'

The experimenter then showed the participants four practice trials presented in the following order:

- inanimate (*flores* 'flowers') presented in the zero cue condition (*tres flores* 'three flowers') like in (5a);
- animate non-gendered (*bebé* 'baby') presented in the one cue condition (on the determiner: *unos bebés* 'some babies') like in (5b);
- animate female (*niña* 'girl') presented in the one cue condition (on the adjective: *niñas simpáticas* 'nice girls') like in (5c);
- animate male (*príncipe* 'prince') presented in the two cue condition (on the determiner and the adjective: *unos príncipes simpáticos* 'some nice princes') like in (5d).

The experimenter made sure the participants understood that they needed to produce the definite article (***el/la***) and the color of the moving picture each time.

The experimenter then asked the participants if they had any questions (*¿Tienes alguna pregunta?*) and started the experimental trials. While showing Figure 1, the experimenter said (7).

(7)   *Ahora vamos a ver…tres noletes*

'Now we are going to see… three noletes'

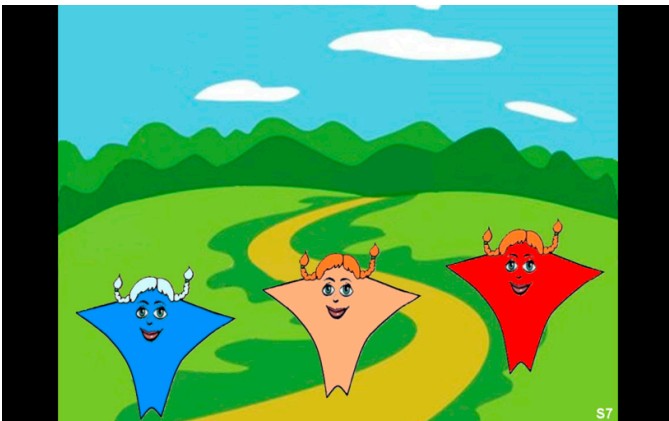

**Figure 1.** Example of animate female gendered characters. *Aquí hay tres noletes* 'here there are three noletes'.

The experimenter then caused one of the objects to move (rotate, move up and down, or the like), and asked (second prompt) (8):

(8)   *¿Cuál se está moviendo?*

'Which one is moving?'

The child was expected to answer either (9a) or (9b):

(9)   a. *el (nolete) rojo*
      b. *la (nolete) roja*

### 2.1.4. Participants

Two groups of Spanish-speaking children participated in the study. The monolingual Spanish (**L1**$_{Sp}$) group consisted of 28 children (11 boys and 17 girls), aged between 5;5 and 6;5 years (Mean = 5;8), who lived in a predominantly monolingual Spanish-speaking region in the area of Pamplona (Navarre) in Spain. Spanish is almost exclusively the language for those children in school, at home, and in their community, though some of their peers living in this area attend other school models where Basque is the language of instruction.

The Basque-dominant bilingual (**BDB**) group consisted of 21 children (13 males and 8 females, aged between 5;5 and 6;4 years (Mean = 5;7) who were growing up in Basque-speaking homes in a Basque-Spanish bilingual area in Ordizia (Gipuzkoa), where most inhabitants use both languages daily. Basque is the main language for those children in school and at home, but they have regular exposure to Spanish in the community.

All participants gave their informed consent for inclusion before they participated in the study. The study was conducted in accordance with the Declaration of Helsinki, and the protocol was approved by the Ethics Committee of the School of Psychology, Bangor University (Project 997).

## 3. Results

Children's responses were coded according to whether participants chose MASC or FEM gender marking on the DP (Determiner and Adjective).

Descriptive data are shown in Table 3. First, there was a strong tendency to produce gender matching DPs: over 95% of the DPs produced by both groups contained the target pseudonoun and Det and Adj modifiers that matched in MASC (10a) or FEM gender (10b,c). Data in parentheses in the examples indicate participant, group and age in year; months.

(10) a.     ***el*** *patibr**o** amarill**o***    (child 4, L1$_{Sp}$, 5;6)

      b.     ***la*** *mardel**a** roj**a***       (child 12, L1$_{Sp}$, 6;0)

      c.     ***la*** *nerte amarill**a***      (child 5, L1$_{Sp}$, 5;11)

Among the remaining items (less than 5% of the sample) the following patterns were attested: Det-Adj gender-mismatching DPs, as in (11a) (such errors were very scarce in the L1$_{Sp}$ group (0.4%) and also in the BDB group (2.98%)); target-deviant pseudonouns such as *espalda* 'back', instead of the pseudonoun *palde,* as in (11b); and target-deviant adjectives such as *gris* 'grey' instead of *negro* 'black', as in (11c). (Such errors occurred less than 2% of the time in both groups.)

(11) a.     ***el*** *(rilape) negr**a***     (child 34, L1$_{Sp}$, 5;5)

      b.     ***la*** *espalda amarill**a***    (child 5, L1$_{Sp}$, 5;11)

      c.     ***el*** *tilent**o** gris*       (child 22, L1$_{Sp}$, 5;11)

There was a clear preference for MASC gender marking across conditions, especially in BDB children (L1$_{Sp}$ = 64%; BDB = 83% of items), as shown in Table 3. (The most prominent response types are shown in bold.)

**Table 3.** Distribution of participants' responses across types (*n* and %).

|  | **L1$_{Sp}$** | **BDB** |
|---|---|---|
| **Masculine Gender (Det = Adj)** <br> **el** (nolete) roj**o** | 856 **(64%)** | 835 **(83%)** |
| **Feminine Gender (Det = Adj)** <br> l**a** (nolete) roj**a** | 419 **(31%)** | 139 (14%) |
| **Mismatch (M/F, F/M)** <br> **el** (nolete) roj**a** | 6 (0.4%) | 30 (3.0%) |
| **Mismatch (Lexical item, stimulus)** | 19 (1.4%) | 0 |
| **Experimenter error** | 13 (1.0%) | 3 (0.3%) |
| **Other** | 31 (2.0%) | 1 (0.1%) |
| **Total** | 1344 (100%) | 1008 (100%) |

Across conditions, absolute numbers and rates of responses related to the variables of referent (male, female, non-gendered animate, artifact), ending (*-o* (MASC), *-a* (FEM), *-e* (non-gendered)) and contextual cues (0 (no gender marking), 1 (gender marking on either Det/Adj), 2 (gender marking on both Det/Adj)). The responses are presented in separated tables.

As shown in Table 4, MASC is clearly preferred over FEM across the four types of referents in both groups, but the lowest rate is found with female animates and the highest with male animates. Ranges of MASC responses across conditions are higher for the BDB group (<81% to 88%>) than for the L1$_{Sp}$ group (<61–74%>).

**Table 4.** Distribution of participants' responses, by referent.

|  |  | **(Biological) Gender of Referents** | | | |
|---|---|---|---|---|---|
|  |  | **Male** | **Female** | **Non-Gendered Animate** | **Artifact** |
| Masculine Responses | L1$_{Sp}$ | 240 **(74%)** | 192 **(61%)** | 215 **(65%)** | 209 **(68%)** |
|  | BDB | 214 **(88%)** | 200 **(81%)** | 213 **(88%)** | 208 **(87%)** |
| Feminine Responses | L1$_{Sp}$ | 84 (26%) | 121 (39%) | 114 (35%) | 100 (32%) |
|  | BDB | 29 (12%) | 48 (19%) | 30 (12%) | 32 (13%) |

Percentages in Table 5 indicate rates of MASC or FEM, out of the total Det__Adj sequences produced by each group by word ending. Across word endings, visible differences in gender assignment were observed in the L1$_{Sp}$ group, shown in Table 5. The only condition in which FEM rates were higher than MASC was with *-a* words (61% vs. 39%). However, such clear differences were attested only for the L1$_{Sp}$ group, since the BDB group revealed a preference for MASC even in the -a ending condition (80%). Both groups preferred MASC with words ending in *-e*, whether they were presented in a MASC or FEM modifier context—the preference is lower for L1$_{Sp}$ group than the BDB group (55% vs. 86%).

**Table 5.** Distribution of participants' MASC form responses, by ending.

|  |  | **N Ending** |  |  |  |
|---|---|---|---|---|---|
|  |  | **-o** | **-a** | **-e (MASC Context)** | **-e (FEM Context)** |
| Masculine | L1$_{Sp}$ | 286 (**89%)** | 124 (39%) | 274 (**86%**) | 172 (**55%**) |
| Responses | BDB | 218 (**88%)** | 194 (**80%**) | 218 (**88%**) | 205 (**87%**) |
| Feminine | L1$_{Sp}$ | 37 (11%) | 196 (**61%**) | 45 (14%) | 141 (45%) |
| Responses | BDB | 29 (12%) | 50 (20%) | 29 (12%) | 31 (13%) |

The distribution of gender marking across numbers of morphological cues provided in the prompts is plotted in Table 6. The percentage of MASC assignment is higher than FEM in all the conditions (0, 1 or 2 cues) in both groups.

**Table 6.** Distribution of participants' responses, by contextual cues.

|  |  | **Number of Morphological Cues on D** |  |  |
|---|---|---|---|---|
|  |  | **0** | **1** | **2** |
|  |  | **No Gender Marking** | **Gender Marking on either D/ADJ** | **Gender Marking on both D/ADJ** |
| Masculine | L1$_{Sp}$ | 308 (**73%**) | 283 (**67%**) | 265 (**62%**) |
| Responses | BDB | 283 (**87%)** | 280 (**86%**) | 272 (**85%**) |
| Feminine | L1$_{Sp}$ | 116 (27%) | 141 (33%) | 162 (38%) |
| Responses | BDB | 43 (13%) | 47 (14%) | 49 (15%) |

Gender responses for pseudowords depending on their semantic and morpho-phonological features are presented in Figures 2 and 3 for both groups. Figures 2 and 3 show the rates of MASC modifiers with pseudonouns ending in *-a*, *-o* (Figure 2), and *-e* (Figure 3) referring to objects and characters with male, female and neutral gender attributes.

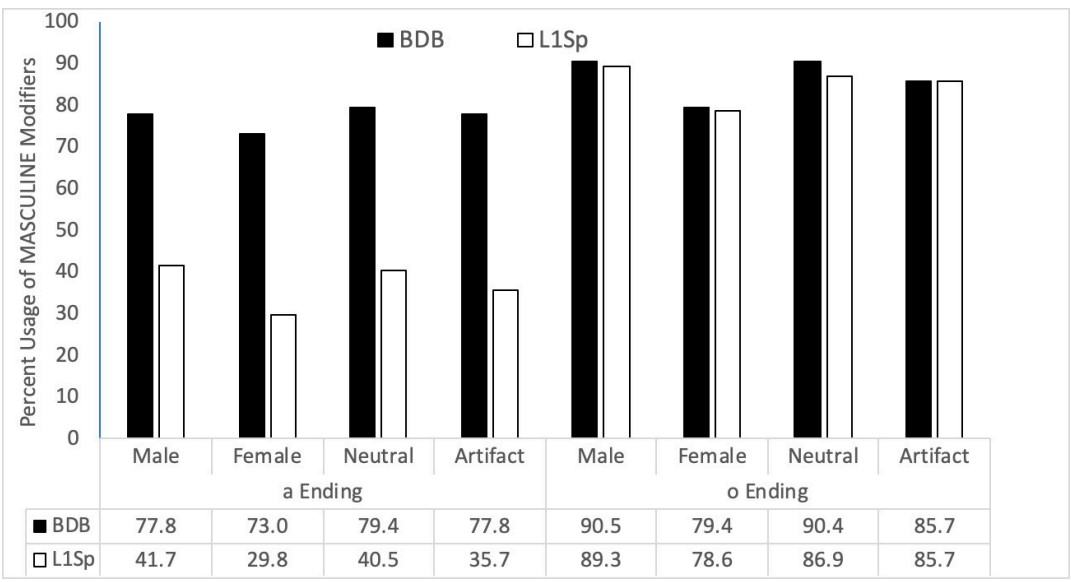

**Figure 2.** Production of MASC modifiers with *-a* vs. *-o* pseudonouns, in reference to male, female, and non-gendered animate objects and artifacts.

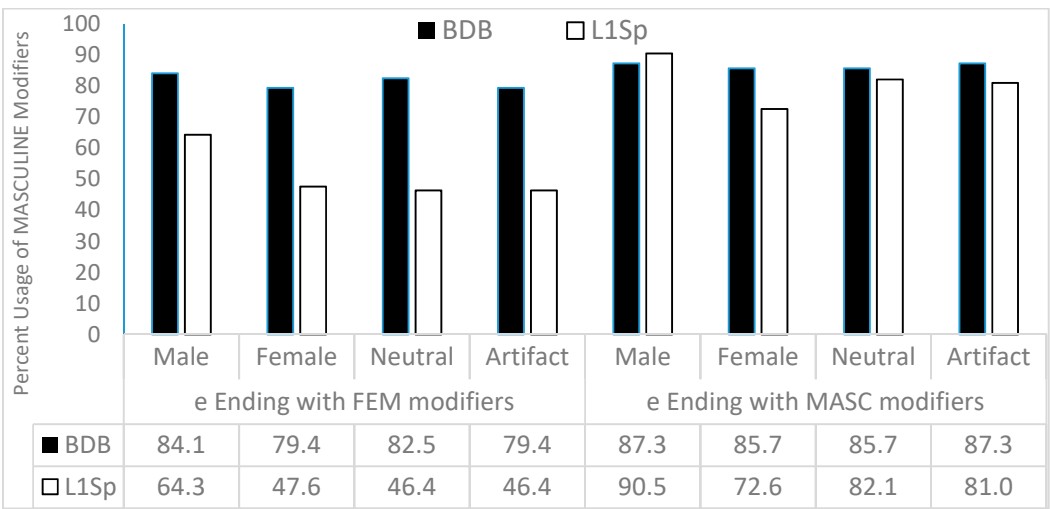

**Figure 3.** Production of MASC modifiers with *-e* pseudonouns in reference to male, female, and non-gendered animate objects and artifacts.

In order to understand these descriptive results, a multiple regression analysis was performed to examine the effects of the different variables on gender assignment to the pseudowords and the relationship between those variables. In these analyses, the MASC was treated as the default, and analyses centered on determining which factors (and to what degree) made participants NOT use a masculine determiner and adjective for a given item. The dependent variable was the percentage of trials for each item in which participants produced masculine determiners and adjectives (e.g., *un nolete rojo* 'a-MASC *nolete* red-MASC') and the independent variables were the ordinal variable NUMBER OF CUES (0, 1, 2 cues), and two categorical predictor variables: REFERENT (Male, Female, non-gendered Animate, Artifact), and WORD ENDING (*-o, -a, -e* with MASC modifiers, *-e* with FEM modifiers).

The categorical predictor variables were dummy-coded in such a way that the male/masculine levels were the point of comparison for the other levels. In the 1-cue and 2-cue conditions, half of the novel words ending in *-e* were paired with masculine cues and half with feminine cues. (Note that, in order to preserve the symmetry of the analysis, half of the novel words ending in *-e* in the 0-cue

condition were also classified as -*e* ending FEM (or "EF") and half as ending -*e* MASC (or "EM"), even if the carrier sentences did not contain gendered determiners or adjectives.)

The table below shows the coefficients for each condition in each group of participants

- Cues: 0, 1, 2
- Ref_F: Female referent
- Ref_N: Non-gendered Animate referent
- Ref_A: Artifact referent
- Ending_A: word with -*a* ending
- Ending_EF: word with -*e* ending accompanied by FEM modifier(s) in the prompt
  Ending_EM: word with -*e* ending accompanied by MASC modifier(s) in the prompt

Results are shown in Table 7. The negative coefficients indicate by how much the dependent variable decreases in the presence of that level of the predictor. Three conditions revealed significance for the two groups: -*a* ending ($p < 0.001$), -*e* ending words with FEM DPs ($p < 0.02$), and female referents ($p < 0.01$). The number of morpho-syntactic cues was significant for the L1$_{Sp}$ group ($p = 0.020$) and at trend level for the BDB group ($p = 0.065$). Thus, for L1$_{Sp}$ speakers, a novel word ending in -*a* makes it 48% less likely that they will assign MASC gender (i.e., use with a MASC determiner/adjective), while for BDB participants the same *a*-ending only reduces MASC responses by 9.5%. Second, novel words ending in -*e* presented in FEM contexts (*aquí hay (unas) noletes (simpáticas)*) are 33.9% and 5.1% less likely to be assigned MASC forms by the L1$_{Sp}$ and BDB group, respectively. Third, female referents are 14.3% (L1$_{Sp}$) and 5.5% (BDB) less likely to be assigned MASC gender, while increasing the number of cues makes it 4.8% less likely to be used with MASC modifiers in the L1$_{Sp}$ group.

**Table 7.** Likelihood of not assigning MASC gender to the pseudoword, across conditions.

|  | BDB | | | L1$_{Sp}$ | | |
|---|---|---|---|---|---|---|
|  | **B** | **Beta ($\beta$)** | **Sig.** | **B** | **Beta ($\beta$)** | **Sig.** |
| Cues | −1.631 | −0.205 | 0.065 | **−4.800** | −0.166 | **0.020** |
| Ref_F | **−5.558** | −0.370 | **0.008** | **−14.300** | −0.262 | **0.003** |
| Ref_N | −0.417 | −0.028 | 0.835 | −7.442 | −0.136 | 0.113 |
| Ref_A | −2.400 | −0.160 | 0.235 | −9.225 | −0.169 | 0.051 |
| Ending_A | **−9.517** | −0.634 | **0.000** | **−48.217** | −0.882 | **0.000** |
| Ending_EF | **−5.142** | −0.343 | **0.013** | **−33.933** | −0.621 | **0.000** |
| Ending_EM | 0.017 | 0.001 | 0.993 | −3.583 | −0.066 | 0.440 |

These statistical analyses indicate that the same factors affect monolingual and bilingual children's assignment of gender to the novel words in Spanish. Five- to six-year-olds show a strong tendency to assign MASC gender to novel words. At the same time, these children show a sensitivity to novel word endings and to the morpho-syntactic context, since -*a* and -*e* ending words presented in FEM DPs show a lower likelihood to get MASC gender assignment. In addition, also significant, but less so than the previous formal properties, is that female referents increase children's tendency to use FEM gender forms with novel words. The referential properties are less relevant than morpho-phonological cues such as word ending and morpho-syntactic cues such as Det-N-Adj agreement in the process of gender assignment to novel words. The finding that the number of morpho-syntactic cues (2 more than 1, and 1 more than 0) decreases the probability of MASC assignment reinforces the preceding statement that children's gender assignment is more dependent on the formal cues provided by the formal properties of words and the grammatical consistency of Spanish morpho-syntax than on the referents' semantic properties.

## 4. Discussion and Conclusions

The current elicitation task elicited gender matching Det-N-Adj structures with novel nouns in order to test monolingual and bilingual children's assignment of MASC/FEM gender to novel words. Interestingly, this experimental procedure requires participants to be sensitive to the Spanish grammatical gender at two levels, the receptive level, as in infant and adult processing studies, and the production level. The design aimed to control participants' responses to overt morpho-phonological and morpho-syntactic gender markers through the inclusion of two independent variables in the prompts: word ending (-*a*, -*e*, -*o*) of novel words and the number (0, 1, 2) of contextual cues. Elicited children's oral responses involved DP-internal gender agreement between the two gender-marked modifiers (the article and the adjective) and the optionally overt novel noun.

All the DPs produced contained an article and an adjective modifying the optionally overt novel N (as in *el (nolete) rojo* or *la (nolete) roja*), which indicates that the procedure is appropriate for testing 5- to 6- year-old children's production of Det_Adj DP structures. The first observation, that over 97% of the DPs produced were gender-matching MASC or FEM structures in both groups, indicates the strength of DP-internal gender-agreement mechanisms in both monolingual and bilingual child Spanish grammars at this age, contrary to the prediction that the BDB group would be less consistent than the L1$_{Sp}$ group.

Secondly, children's production of elicited Spanish DPs based on prompts containing novel words reveal that Spanish-speaking children have a strong tendency to assign MASC gender to novel words, not only to those with male referents but to non-gender specific and inanimate referents, as well as to words with female referents. This is especially true for the BDB children (83% MASC), but also applies to the L1$_{Sp}$ children (63% MASC). The overuse of MASC gender is widely attested across studies on monoglot adult and child Spanish, such as the elicited production of Det and Adj with real words by bilinguals, where bilinguals showed more overuse (Cuza and Pérez-Tattam 2015). However, an important methodological difference between the two types of studies (real words vs. novel words) should be highlighted. As mentioned in the introduction, one of the advantages of experimental tasks using novel words is that they allow the researcher to control the effect of being able to access (target) gender features through lexical knowledge of the real words. Such effect disappears with novel words and allows us to discover the relevant (semantic, lexical, syntactic) strategies for gender marking in bilingual as compared to monolingual children on a level playing field. Thus, the current experimental setting simulates quite accurately the situation in which children are exposed to the use of unknown (real or not) words in their natural use of Spanish. The generalized use of MASC has been also attested in many studies on mixed DPs containing a Spanish Det and the lexical insertion of an English N (Liceras et al. 2008; Valdés Kroff et al. 2017) or of a Basque N (see Badiola and Sande 2018 but Couto et al. 2016).

This extended use of MASC could be considered to be compatible with some MASC default strategy. Nevertheless, the results of this study confirm that 5- to 6-year-old children's gender assignment is affected by the morpho-phonological (ending) and syntactic (agreement) properties of novel words in two ways: first, the children in this study produced different rates of MASC modifiers with words ending in -*e* when these were presented in FEM agreement contexts than with words ending in -*e* when they were presented in MASC agreement contexts. Secondly, the number of morpho-syntactic cues affected the likelihood of participants assigning MASC gender to novel words.

Rates of MASC forms in the DPs elicited were generally higher than the rates of FEM in the case of most endings, but those MASC rates were less likely to be found with words ending in -*a* and words ending in -*e* when accompanied by FEM marked (Det)_(Adj) prompts. Thus, data indicate that 5- to 6-year-olds rely on formal properties of the language (word ending and DP internal agreement) to assign the gender features to novel nouns as an effect of their Spanish grammar's gender agreement mechanisms. Such consistency has been attested with infants, who are able to associate novel words' gender features with their endings and with their modifiers' gender marking (Arias-Trejo and Alva 2013; Arias-Trejo et al. 2013).

Thus, the first prediction that all children will assign gender to novel words based on semantic (referential) and formal (morpho-syntactic and morpho-phonological) cues and that children with

regular exposure to the Spanish language will produce consistent DP internal agreement has been partially confirmed, as long as no clear prediction is made regarding the prevalence of some cues over others.

The results obtained are also partially consistent with the second prediction, that 5-year-old BDB children will differ from their L1$_{Sp}$ monolingual peers in that they are less constrained in their gender assignment to novel words, a pattern which is compatible with the stronger preference for MASC assignment observed in the BDB group.

With regard to the factors that influenced children to deviate from this general preference for MASC forms in favor of FEM forms, both groups of children were significantly influenced by the following three factors: word ending in *-a*, occurrence with FEM modifiers, and presence of a female referent. The L1$_{Sp}$ children were also significantly influenced by the number of morphological cues (with more cues more likely to lead to application of FEM forms in the response), but the BDB children also came to a near-significant level in being influenced by the number of cues.

a. For the L1$_{Sp}$ children, the *-a* word ending was most influential ($\beta$ = |0.88|), followed by occurrence (for *-e* ending words) in a FEM context (either preceded by a FEM article and/or followed by a FEM-marked adjective) ($\beta$ = |0.62|), and then more distantly followed by having a female referent ($\beta$ = |0.26|), and having an increased number of cues (article plus adjective > only article or adjective > no cues) ($\beta$ = |0.17|).

b. For the BDB children, the hierarchy was similar, except that the influence of the occurrence in the context of FEM modifiers was lower, and the impact of a female referent was more elevated: for these children, the *-a* word ending was, as for the L1$_{Sp}$ children, the most influential ($\beta$ = |0.63|); the presence of FEM modification (with *-e* ending words) was less influential than for the L1$_{Sp}$ children ($\beta$ = |0.34|); and having a female referent was slightly more influential ($\beta$ = |0.37|). Having an increased number of cues ($\beta$ = |0.21|) was near-significant ($p$ = 0.65).

These combined results suggest, on the whole, similar gender assignment processes in the L1$_{Sp}$ and BDB children: MASC preference is revealed for both groups, while at the same time children show sensitivity to both formal and referential properties of novel nouns. The data demonstrate that, regardless of children's stronger language, the MASC is the unmarked option. However, all these children pay attention to word endings and to Det-N-Adj agreement, since for the two groups words ending in *-a* are less likely to elicit MASC agreement. This is in line with the assumption that FEM gender is the marked option in Spanish, predominantly attested in words ending in *-a*, rarely associated with words ending in *-o*, and possible with other endings such as words ending in *-e* (Harris 1991). It is precisely with these phonological forms that children are less prone to assign MASC and to look for additional cues, either in the morpho-syntactic context or in the gender properties of animate referents. Contrary to what might have been predicted, referential (feminine gender) properties of the novel words with animate referents were less determinant than formal properties such as word ending alone (*-a*) or together with morpho-syntactic cues. These both seem to provide reliable cues for assigning the more marked FEM gender.

Both groups show, then, that gender assignment for novel nouns is processed separately from a word's lexico-semantic content. However, the type of exposure to the grammatical gender language (here Spanish) and the lack of grammatical gender in the dominant language (here, Basque) of the BDB group can influence the extent to which the female gender of referents is taken into account in the assignment of FEM gender. Another consideration, as pointed out by one reviewer, is the fact that the BDB children speak a language in which final *-a* marks something other than gender (absolutive singular in Basque). This might make the form-function mapping for the BDB children particularly opaque. In other words, this could make the gender marking less "transparent" for bilingual than for monolingual Spanish-speaking children.

These results are consistent with those reported in similar work showing both effects of input and influence from a non-gendered language (Gathercole 2002; Gathercole et al. 2001; Lemmerth and Hopp 2017; Munarriz et al. 2018; Unsworth et al. 2014). They provide additional evidence that even when the language in which grammatical gender occurs is the dominant language of the community such input and inter-language effects may accrue.

The fact that both groups show a strong tendency to assign MASC gender to novel words as well as a strong consistency for morpho-syntactic agreement and that they appear to apply the same hierarchy of cues driving non-default gender assignment is especially relevant. The commonality of response types suggests that a mutual hierarchy may apply to (at least) regular users of the Spanish language regardless of their monolingual or bilingual linguistic profile.

The results obtained from unbalanced Basque-Spanish bilingual children are compatible with the feature re-assembly approach (Lardiere 2008; Putnam and Sánchez 2013) proposed for sequential (often heritage) bilingual children. According to this hypothesis, the levels of activation of the lexicon and the strength of association between functional, semantic, and phonological features would be lower in the bilinguals' than in the monolinguals' Spanish grammar. Nevertheless, the strength of the phrase-internal agreement and the similar hierarchy of cues, in addition to the strong preference for MASC observed in both groups, indicates that the level of association of most features is strong for this particular group. The materials used (pseudowords instead of real words) and the experimental design (elicited production and prompting of gender assignment using 0, 1, 2 contextual cues) limits testing the level of activation of the lexicon.

Finally, it should be recalled that the two groups of children differ in the amount of exposure to and the degree of use of the Spanish language. They probably also differ in their age of acquisition (since most children of the BDB group are offspring of two Basque-speaking parents, even though they live in an area where Spanish is present alongside Basque in social interactions). The similarities found between the two groups of children growing up in a longstanding language contact situation are in line with Gathercole and Thomas (2009), who concluded that children growing up in stable bilingual communities "may all acquire the dominant language to equivalent levels regardless of home language patterns" (p. 233).

In future studies, it would be interesting to explore the role of other inflectional processes (such as number inflection) in gender assignment by Basque-Spanish bilinguals. As pointed out by one reviewer, the results might be different with plural forms of the definite article or with the demonstrative, which have explicit gender marking in the masculine forms (definite article *los* MASC pl.; demonstratives *este* MASC sg./*estos* MASC pl.). Including plural nouns would allow testing for the interaction between gender assignment and other inflectional processes. In addition, as pointed out by one reviewer, the *-o* = MASC and *-a* = FEM "rule" in Spanish seems exceptionally strong. It would be interesting more generally to conduct further research into the questions addressed by this study in languages with three genders (e.g., German or Russian), or with two genders but less clear-cut strategies for gender assignment than Spanish (e.g., Dutch, Swedish/Norwegian/Danish).

**Author Contributions:** Conceptualization, V.C.M.G., R.P.-T., H.S.-G., M.J.E.; Methodology, V.C.M.G., H.S.-G., R.P.-T., M.J.E.; Software, H.S.-G.; Validation, R.P.-T., H.S.-G., V.C.M.G.; Formal Analysis, H.S.-G., V.C.M.G.; Investigation, M.J.E.; Resources, M.J.E., H.S.-G., V.C.M.G.; Data Curation, M.J.E., R.P.-T., H.S.-G., V.C.M.G.; Writing-Original Draft Preparation, R.P.-T., M.J.E., V.C.M.G.; Writing-Review & Editing, V.C.M.G., M.J.E., R.P.-T.; Visualization, R.P.-T., V.C.M.G.; Supervision, V.C.M.G., M.J.E.; Project Administration, M.J.E.; Funding Acquisition, M.J.E., V.C.M.G.

**Funding:** This research was funded by the Basque Government (IT983-16) & MINECO/FEDER (FFI2015-68589-C2-1-P) and by the ESRC Centre for Bilingualism, Bangor, UK.

**Acknowledgments:** To the children and to the schools who kindly agreed to participate in our study. To Tania Barberán and Isabel García del Real, whose assistance in collecting the data is greatly appreciated. This study was supported by the Basque Government (IT983-16) & MINECO/FEDER (FFI2015-68589-C2-1-P) and by the ESRC Centre for Bilingualism.

**Conflicts of Interest**: The authors declare no conflicts of interest.

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
