# Peer review of "Gender Assignment to Spanish Pseudowords by Monolingual and Basque-Spanish Bilingual Children"

_languages, doi:10.3390/languages4030058_

Round 1

Reviewer 1 Report

This is a well written and well documented paper on a classical topic in the study of grammatical gender assignment from the perspective of bilingual speakers. The methodology is well suited for the research questions and the conclusions are congruent with the results. The presentation, structure, style and grammar are impeccable. The author(s) seem(s) highly familiar with the subject, which is reflected in the presence of several abbreviations which are surely obvious to the author(s) but which would merit a more explicit treatment for readers which are not that familiar with this area of research (e.g. AoA, cL2, ESRC). 

Overall this is a scientifically sound paper presenting relevant results on an important topic. I thus recommend it for publication.

On a general level, it would be interesting, however, to see more research into the questions addressed here on languages which have less clear-cut strategies for gender assignment than Spanish (where the -o = MASC and -a = FEM "rule" is very strong). What happens, for example, in languages with three genders (e.g. German or Russia), or with two genders but which are much more arbitrary than Spanish el, la (e.g. Dutch, Swedish/Norwegian/Danish)?

I attach the pdf-document with my detailed comments and questions (which mainly have to do clarity of expression) should the author(s) want to have a look at them.

Author Response

Response to Reviewer 1 Comments

This is a well written and well documented paper on a classical topic in the study of grammatical gender assignment from the perspective of bilingual speakers. The methodology is well suited for the research questions and the conclusions are congruent with the results. The presentation, structure, style and grammar are impeccable.

Thank you for your constructive feedback – it is good to know that these aspects of the paper are satisfactory.

The author(s) seem(s) highly familiar with the subject, which is reflected in the presence of several abbreviations which are surely obvious to the author(s) but which would merit a more explicit treatment for readers which are not that familiar with this area of research (e.g. AoA, cL2, ESRC).

Response 1: Thank you for your observation. Another reviewer has picked this up too. These have now defined these on the first use.

On a general level, it would be interesting, however, to see more research into the questions addressed here on languages which have less clear-cut strategies for gender assignment than Spanish (where the -o = MASC and -a = FEM "rule" is very strong). What happens, for example, in languages with three genders (e.g. German or Russia), or with two genders but which are much more arbitrary than Spanish el, la (e.g. Dutch, Swedish/Norwegian/Danish)?

Response 2: These interesting questions to be taken on board in future research. These have been added to the Discussion.

Could number inflection play a role? I mean: Do plural inflected nouns show the I attach the pdf-document with my detailed comments and questions (which mainly have to do clarity of expression) should the author(s) want to have a look at them.

Response 3: Please see attached document.

Reviewer 2 Report

This is an interesting paper on the acquisition and status of gender in Spanish. The research is well designed and the results are pertinent for both theoretical grammar and language acquisition studies. The fact that grammatical cues seem to be more pertinent than semantic or referential cues is interesting for the analysis of the DP structure and for the need to emphasize the role of formal features and syntactic relation in the language acquisition process. The parallelism between bilingual and non-bilingual learners is also worthy and it is very relevant for future research (until what extent does the parallelism hold? Which grammatical features/relations/contexts are favored (or unfavored)? etc.).

In addition to these general remaks I have a few comments/suggestions (it's up to the author to incorporate them; maybe s/he already thought about it or consider them useful for future research):

1) In Basque nouns often appear with a final '-a' that "corresponds" to the Spanish definite article (nothing to do with gender marking). Might this intervene on the non-consideration of final '-a' as a feminine marker in Spanish words (by BDB speakers)? Probably, most of the contexts in the test involve an absolutive singular form (the one with final '-a'). Could it be the case that other forms (singular or plural) intervene in gender assignment?

2) Could number inflection play a role? I mean: Do plural inflected nouns show the same behavior concerning gender assignment? I think that all examples were in singular (maybe I'm wrong). I say this because the singular definite article form 'el' has no explicit gender marking, in contrast with 'la' or 'los'. It may also be interesting to check whether the results are the same with a demonstrative (other contexts would be required, obviously), which appears with a final '-e' in masculine. In addition, paying attention to plural nouns would allow to test the possible influence of the interaction with another inflectional process intervenes.

3) The Discussion and Conclusion section could be enriched/improved with a comparison with English/Spanish learners. This would be more sense to the remarks in section 2 and the whole paper would gain in consistency.

There are also some remarks concerning format, typos ...:

- I suggest to give the full form of AoA in line 47. I think it has not been given before. 

- Line 159. I think that 'additionally' should be 'additional'

- Line 221. A paragraph that consists of only one short sentence is odd.

- Table in line 303. I counted 26 monosyllabic forms and 24 bisyllabic forms. It is not exactly half and a half (though it is certainly very very close). 

- Table in line 316. Format of 'Version A' (upper left cell).  

- Line 353. Is 'Figure 2' correct?

- Line 453. Delete one 'the'?

Author Response

Response to Reviewer 2 Comments

This is an interesting paper on the acquisition and status of gender in Spanish. The research is well designed and the results are pertinent for both theoretical grammar and language acquisition studies. The fact that grammatical cues seem to be more pertinent than semantic or referential cues is interesting for the analysis of the DP structure and for the need to emphasize the role of formal features and syntactic relation in the language acquisition process. The parallelism between bilingual and non-bilingual learners is also worthy and it is very relevant for future research (until what extent does the parallelism hold? Which grammatical features/relations/contexts are favored (or unfavored)? etc.).

Thank you for your constructive feedback – it is good to know that the application to both theoretical grammar and language acquisition is clear in this paper.

In addition to these general remarks, I have a few comments/suggestions (it's up to the author to incorporate them; maybe s/he already thought about it or consider them useful for future research):

1) In Basque nouns often appear with a final '-a' that "corresponds" to the Spanish definite article (nothing to do with gender marking). Might this intervene on the non-consideration of final '-a' as a feminine marker in Spanish words (by BDB speakers)? Probably, most of the contexts in the test involve an absolutive singular form (the one with final '-a'). Could it be the case that other forms (singular or plural) intervene in gender assignment?

Response 1: This is a very interesting remark. The fact that the BDB children are bilingual, and speak a language in which final -a marks something other than gender (absolutive singular) might make the mapping for them particularly opaque.  In other words, this could make the gender marking less "transparent" for them than for monolingual Spanish-speaking children. We have added this to the Discussion (L 632).

2) Could number inflection play a role? I mean: Do plural inflected nouns show the same behavior concerning gender assignment? I think that all examples were in singular (maybe I'm wrong). I say this because the singular definite article form 'el' has no explicit gender marking, in contrast with 'la' or 'los'. It may also be interesting to check whether the results are the same with a demonstrative (other contexts would be required, obviously), which appears with a final '-e' in masculine. In addition, paying attention to plural nouns would allow to test the possible influence of the interaction with another inflectional process intervenes.

Response 2: This is a good point and something we would consider exploring in a follow-on study. We have added these questions at the end of the discussion on questions for future study.

3) The Discussion and Conclusion section could be enriched/improved with a comparison with English/Spanish learners. This would be more sense to the remarks in section 2 and the whole paper would gain in consistency.

Response 3: This is a good idea that would definitely add depth to the discussion. At this time, those data are not well enough analysed to be able to add such a discussion within the period given by the editors, but we will consider taking the idea on board for future publications.

There are also some remarks concerning format, typos ...:

- I suggest to give the full form of AoA in line 47. I think it has not been given before.

Done. AoA changed to Age of Acquisition (AoA).

- Line 159. I think that 'additionally' should be 'additional'.

Done. ‘Additionally’ changed to ‘Additional’.

- Line 221. A paragraph that consists of only one short sentence is odd. 

They should be part of the same paragraph. Have left a note for the copy-editor for it to be changed.

- Table in line 303. I counted 26 monosyllabic forms and 24 bisyllabic forms. It is not exactly half and a half (though it is certainly very very close).

Done. Changed to “approximately half were monosyllabic, half were disyllabic”.

- Table in line 316. Format of 'Version A' (upper left cell). 

Have left a note for the copy-editor. It was fine in the original word submission.

- Line 353. Is 'Figure 2' correct? 

Fixed. Changed to “Figure 1”.

- Line 453. Delete one 'the'?

Deleted.

See also corrections in attachment.
